# Therapeutic Strategies for Targeting IL-1 in Cancer

**DOI:** 10.3390/cancers13030477

**Published:** 2021-01-26

**Authors:** Adrian Gottschlich, Stefan Endres, Sebastian Kobold

**Affiliations:** 1Center for Integrated Protein Science Munich (CIPS-M) and Division of Clinical Pharmacology, Department of Medicine IV, University Hospital, Ludwig-Maximilians-Universität München, 80337 Munich, Germany; adrian.gottschlich@med.uni-muenchen.de (A.G.); Stefan.Endres@med.uni-muenchen.de (S.E.); 2German Center for Translational Cancer Research (DKTK), Partner Site Munich, 80337 Munich, Germany; 3Einheit für Klinische Pharmakologie (EKLiP), Helmholtz Zentrum München, German Research Center for Environmental Health (HMGU), 85764 Neuherberg, Germany

**Keywords:** IL-1-blockade, clinical trials, cancer, immunotherapy, adoptive T cell therapy, chimeric antigen receptor T cells, CAR

## Abstract

**Simple Summary:**

Interleukin-1 cytokines are key proinflammatory cytokines which have been implicated with differing pro- and antitumorigenic properties. Recent years have brought exciting insights and developments in IL-1-targeted therapies. Here, we present an overview of past and present research focusing on the role of IL-1 in cancer, with a special focus on clinical research and on therapeutic implications. With this, we strive to assist scientists in their future research objectives and to highlight possible directions for IL-1-targeting therapies in the coming years.

**Abstract:**

Since its discovery, interleukin-1 has been extensively studied in a wide range of medical fields. Besides carrying out vital physiological functions, it has been implicated with a pivotal role in the progression and spreading of different cancer entities. During the last years, several clinical trials have been conducted, shedding light on the role of IL-1 blocking agents for the treatment of cancer. Additionally, recent developments in the field of immuno-oncology have implicated IL-1-induced signaling cascades as a major driver of severe chimeric antigen receptor T cell-associated toxicities such as cytokine release syndrome and immune effector cell-associated neurotoxicity. In this review, we summarize current clinical trials investigating the role of IL-1 blockade in cancer treatment and elaborate the proposed mechanism of these innovative treatment approaches. Additionally, we highlight cutting-edge developments utilizing IL-1 blocking agents to enhance the safety and efficacy of adoptive T cell therapy.

## 1. Introduction

Interleukin-1 (IL-1), then amongst other things named human leukocytic pyrogen, was first described in 1974 by Dinarello et al. [1]. In 1984 and 1985, the amino acid sequence of two distinct peptides were published, hence known as interleukin-1α (IL-1α) and interleukin-1β (IL-1β) [2,3]. Over the next decades, a nearly unprecedented interest in the field of cytokine biology was spiked, paving the way to unravel the involvement in many fundamental signaling cascades in different areas of biomedical research.

While the involvement of the IL-1 cytokines and their counterplayers in the development, progression and metastasis of cancer has been described in preclinical models for decades, it was not until 2015 that Ridker et al. were able to provide more solid evidence in a clinical setting. In the Canakinumab Anti-inflammatory Thrombosis Outcomes Study (CANTOS), the authors observed, as a prespecified safety analysis, a dose-dependent decrease in overall cancer mortality and in the incidence of non-small-cell lung cancer (NSCLC) in patients treated with an anti-IL-1β antibody [4]. While merely being a proof-of-concept study and with the underlying mechanisms far from being understood, the concept itself is intriguing, opening the door for a new approach to cancer immunotherapy. 

Rébé and Ghiringhelli recently published an extensive review about the pleiotropic functions of interleukin-1β in cancer [5]. While in general, most reports focus on the protumoral effects of IL-1β in cancer biology, the authors highlighted antitumoral functions of IL-1β as well, raising important considerations for future developments of IL-1 blocking strategies in cancer. Figure 1 recapitulates the most important pro- and antineoplastic functions of IL-1β.

Building on their review, we give an overview of clinical trials currently being conducted assessing the potential of IL-1 blockade in cancer. Furthermore, we will summarize innovative recent developments of combining IL-1 blocking agents with adoptive T cell therapy (ACT) to improve the efficacy and safety of these cell therapeutic approaches.

## 2. Therapeutic Targeting of IL-1 in Cancer

In preclinical models of hematological and non-hematological malignancies, the antitumoral effects of IL-1β have been investigated for decades. In the 1990s, for example, investigators were able to demonstrate reduced number and size of metastases in murine and human melanoma models through administration of an IL-1 receptor antagonist (IL-1Ra) [6,7,8]. Furthermore, in 2003, Voronov et al. reported on the proangiogenic functions of IL-1, with blockage leading to a reduction in tumor growth of murine melanoma and breast cancer models in vivo [9,10]. As mentioned, the pleiotropic functions of IL-1β in cancer immunity have recently and extensively been reviewed [5,11,12,13,14,15,16]. Thus, we will not further elaborate the underlying mechanisms, but focus on past and present clinical trials assessing the role of IL-1 blockade in the treatment of malignant disease.

### 2.1. Targeting IL-1 in Hematological Malignancies 

One of the earliest trials, which aimed to investigate the role of modulating the IL-1 axis in cancer patients, was conducted at the Mayo Clinic in Rochester, Minnesota [17]. Patients suffering from multiple myeloma (MM), or more precisely, from its benign precursor conditions, were either treated with the IL-1 receptor antagonist (IL-1Ra) anakinra alone or in combination with dexamethasone (NCT00635154). MM was chosen for two major reasons: (1) preclinical studies were able to identify the pivotal function of MM cell-derived IL-1β to induce IL-6 production in stromal cells, which in turn acts as an essential growth factor for malignant myeloma cells [18,19,20,21]; and (2) there are well-defined precursor conditions such as monoclonal gammopathy of undetermined significance (MGUS), smoldering multiple myeloma (SMM) and indolent multiple myeloma (IMM), which, however, depending on the risk profile of each individual patient, can progress and develop into life-threatening and often deadly disease [22,23,24]. As, due to the lack of effective treatment regimes for these precursor conditions, current clinical guidelines recommend a watch-and-wait strategy after diagnosis, especially patients at high risk for disease progression would greatly benefit from innovative therapeutic options [25]. Thus, authors initiated a clinical trial to assess the role of the blockade of the IL‑1–IL‑6 axis in benign precursors of MM and its influence on the subsequent progression to malignant disease.

Forty-seven eligible patients were treated with either anakinra alone or in combination with dexamethasone. High-sensitivity C-reactive protein (hs-CRP) was used as an indicator of successful inhibition of the IL-1–IL-6 axis, as both IL-1 and IL-6 are known to promote acute-phase reactions in the liver [26,27]. All 47 patients received IL-1Ra, while 25 of 47 additionally received dexamethasone. Dexamethasone was administered based on preclinical data, illustrating a synergistic treatment effect in combination with anakinra [20]. Overall, 25 of 47 (responders) patients showed a decrease in their baseline hs-CRP, indicating successful targeting, while 22 of 47 (non-responders) did not. Responders exhibited a significantly improved overall (OS) and progression-free survival (PFS). Thus, investigators were able to give clinical validation for their preclinical research and were able to highlight the importance of the IL‑1–IL-6 axis in the progression of MM. Currently, the same study group is conducting a phase I trial combining IL-1Ra treatment with dexamethasone and lenalidomide for the treatment of early stages of MM (NCT02492750).

Besides directly targeting the IL-1–IL-1 receptor (IL-1R) axis, therapeutic modulation of downstream signaling using protein kinase inhibitors has also been investigated. As such, constitutive activation of the IL-1R pathway and its fundamental signal transducer proteins (interleukin-1 receptor-associated kinase 1, IRAK1; myeloid differentiation primary response 88 protein, MYD88) has been demonstrated in several hematological malignancies [28]. MYD88 is an adaptor protein, which is recruited to the IL-1R or Toll-like receptor (TLR) complexes following activation, leading to the phosphorylation of IRAK1. Through different mechanisms, IRAK1 ultimately leads to an activation of different proinflammatory pathways (e.g., nuclear factor kappa-light-chain-enhancer of activated B cells (NF-κB)), delivering important proliferative signals to the malignant cells [29]. Constitutive signaling of IRAK1 has, for example, been observed in T cell acute lymphoblastic leukemia (T-ALL) [30] and myelodysplastic syndrome (MDS) [31]. Similarly, activation mutations in the MYD88 protein are observed in up to 95% cases of Waldenström macroglubinemia [32]. Recently, investigators were able to demonstrate the potential of IRAK1-targeted small-molecule inhibition in acute myeloid leukemia (AML) as well [33].

### 2.2. IL-1 Blockade in Solid Tumors 

The IL-1 cytokines have not only been implicated in the development and progression of hematological malignancies, but have also been attributed with increased incidence [34,35], poor prognosis [36], rapid disease development [8] and high metastasis burden [7,8] in different solid tumor entities. As such, the potential of blocking the IL-1 axis in cancer treatment was recognized early on [17] and several clinical trials investigating the combination of IL-1 blockade with standard chemotherapy-based treatment regimes in several solid malignancies were initiated.

One of the first trials testing IL-1 blockade in solid tumors was conducted between 2010 and 2012 at the MD Anderson Cancer Center (NCT01021072). In an open-label phase I trial, investigators tested MABp1 (Xilonix), a fully humanized antibody targeting interleukin-1α, in 52 patients with advanced cancer. MABp1 was well tolerated and no dose-limiting toxicities were observed during this study (primary endpoint). Secondary endpoints included antitumor activity and disease control. Here, significant improvement in lean body mass could be observed in some of the treated patients [37]. Interestingly, a subgroup analysis of patients suffering from NSCLC revealed a (non-significant) trend towards a prolonged median survival in the MABp1-treated group [38].

In the phase III randomized, double-blind, placebo-controlled follow-up trial (NCT02138422), patients suffering from advanced colorectal cancer (CRC) were treated with MABp1. The combined primary endpoint (composite of lean body mass and improvement in the cancer-related symptoms of pain, fatigue and anorexia) was more often reached in the treatment group [39]. However, since the primary endpoint was chosen as a composite parameter and the cohort size was calculated on this basis, the study was not able to determine benefits in overall survival [39]. Accordingly, another phase III study with a larger patient cohort was initiated (NCT01767857), which was however terminated as the prospective futility boundary of the primary endpoint (overall survival) was met. In summary, while blockade of IL‑1α seems to be effective in reducing inflammation-associated symptoms such as cachexia or other cancer-related symptoms (pain, fatigue, anorexia) in patients with advanced malignant disease, a benefit in overall survival has so far not been shown. These concerns, amongst others, lead to a refusal of the marketing authorization by the European Medicines Agency (EMA).

However, IL-1α and IL-1β are distinct cytokines with differing functions [40]. Functional disparities between IL-1α and IL-1β blockade in preclinical cancer models have been observed [41]. Accordingly, the potential of solely blocking IL-1β in cancer treatment needs to be carefully evaluated in the clinical setting as well.

In 2017, Ridker et al. published additional findings (the primary endpoint of the study was nonfatal myocardial infarction, nonfatal stroke or cardiovascular death) of the multicentric, randomized, double-blind CANTOS trial [4,42]. The study was designed to assess the benefit of IL‑1β blockade through application of the anti-IL-1β antibody canakinumab as a secondary prevention strategy in patients with cardiovascular disease. Inflammation itself, but also the IL-1‑induced signaling cascade, has been implied as an important contributor to the development of atherosclerotic lesions [43,44,45,46]. However, successful prevention strategies targeting this mechanism are still missing in clinical routine [47]. To identify high-risk patients who could benefit from anti-inflammatory prevention strategies, only patients with elevated hs-CRP levels (above 2 mg/L) were included in the study. Surprisingly, besides differences in the primary endpoints, the authors also observed a dose-dependent reduction in total cancer mortality and in the incidence of lung cancer following application of canakinumab. In an effort to deliver more detailed insights into the molecular characteristics of the developed malignancies, investigators recently analyzed circulating tumor DNA (ctDNA) of the collected plasma samples of the CANTOS cohort [48]. Treatment with canakinumab did not result in the enrichment of a specific molecular lung cancer subtype, suggesting its therapeutic potential across these different subgroups. Interestingly, when examining baseline ctDNA (plasma samples before treatment) of patients who developed lung cancer over the course of the study, common cancer mutations (Catalogue of Somatic Mutations in Cancer database, COSMIC) were detected in 43% of patients who later developed cancer (29/67). In comparison, COSMIC mutations were only found in 20% of patients who did not develop cancer over the course of the study (4/20) [48].

While these results are intriguing and can potentially open the door for future clinical applications, two major points need to be addressed: (1) As mentioned, the primary purpose of the CANTOS trial was the investigation of a new prevention strategy in patients with cardiovascular disease. As such, study design, patient cohorts and treatment schedules were not specifically tailored to the detection of newly developed cancers or the treatment of such. Cancer incidence, however, was a prespecified safety parameter of the study. While, due to the randomized nature of the trial, patient characteristics should be equally distributed, important cofounders, such as undetected malignancies, cannot be ruled out with certainty. (2) To date, the reasons for the observed effects in the CANTOS trial are not well understood. As described by Mantovani et al. [11], one is tempted to speculate that blocking of the IL-1 axis in these patients leads to a reduction of the immunosuppressive capacities of both myeloid-derived suppressor cells (MDSC) and tumor-associated macrophages (TAM), as it has been shown in different preclinical models [49,50,51]. However, experimental proof of this hypothesis is missing. Arising from these considerations, an attractive therapeutic strategy includes a combinatorial approach, combining checkpoint inhibition with anti-IL-1 blocking agents. This rationale is also strengthened by results from preclinical models, in which combination of anti-IL-1β blocking antibodies with programmed cell death protein 1 (PD-1) blockade exhibited significantly improved tumor control and overall survival of mice in murine breast cancer models [51]. In summary, both preclinical and clinical investigations are needed with the aim to validate these findings in the clinical setting, to unravel the underlying mechanism which contributed to the reduced incidence of malignant disease and to identify suitable treatment strategies for the application of IL-1 blocking agents in different cancer entities.

Sparked by the findings in the CANTOS trial, a global clinical trial program—the Canakinumab Outcomes in Patients with NSCLC Study (CANOPY)—has been initiated. Four different trials (CANOPY 1, 2 and CANOPY-N, A) are evaluating distinct treatment regimens in different stages of disease (NCT03631199, NCT03626545, NCT03968419, NCT03447769). In addition, a multiplicity of other combinatory strategies in several cancer entities are currently under investigation. Table 1 gives an overview of the clinical trials currently registered at clinicaltrials.gov, assessing the potential of IL-1 blockade alone or in combination with other drugs in the treatment of cancer.

### 2.3. Safety of IL-1 Blocking Agents

Addition of new drugs to existing therapeutic regimes requires careful evaluation of the risk–benefit relations. Anti-inflammatory treatment strategies often come with an increased risk for serious and opportunistic infections. As such, treatment with tumor necrosis factor alpha (TNF‑α) blocking agents increases the risk of tuberculosis, *Pneumocystis jirovecii* infection and aspergillosis, and meta-analyses have reported increased risk for serious infectious complications [53,54]. While common adverse effects of IL-1 blocking agents include reduction in leukocyte and thrombocyte counts, the overall safety profile is excellent [55]. For anakinra, the most common side effects are injection site reactions such as redness, itching or rash (generally, anakinra is subcutaneously administered once daily). The overall rate of reported adverse effects are low, with most side effects being mild. In patients with cryopyrin-associated periodic syndromes (CAPS), a hereditary inflammatory disorder with gain‑of‑function mutations of the NLR family pyrin domain containing 3 (NLRP3) inflammasome, Kullenberg et al. for example published long-term safety profiles of patients treated with anakinra. Here, overall rates of adverse effects were less than 0.8 events per patient year and no differences amongst different age groups (infants, <2 years; children, 2–11 years, adults ≥18 years) were observed [56]. Treatment with canakinumab during the CANTOS trial did reveal significantly higher incidence rates of infection and sepsis [42]; however, in absolute numbers, treatment in over 30,000 patient‑years resulted only in an excess of 1.3 fatal infections per 1000 patient years [46]. Although thrombocytopenia was more common in the canakinumab-treated groups, no increased risk for hemorrhage was observed. Additionally, no differences in incidence rates of opportunistic infections were registered [42].

In summary, while the overall safety profile of IL-1 blocking agents are rather favorable, the reduction in overall leukocyte and, most importantly, neutrophil counts need to be taken into account. With an overall increased risk for fatal infections, future studies need to include well-planned monitoring and detection strategies for infectious complications [4]. Along these lines, determined initiation of anti-infectious treatment will be crucial to prevent serious adverse effects, especially in elderly patients.

### 2.4. Novel Treatment Strategies to Inhibit IL-1-Mediated Signaling

As IL-1β undergoes a distinct processing and secretion mechanism [57,58], modulation of the IL-1 axis can not only be achieved through conventional strategies (receptor antagonist, receptor-blocking antibodies, neutralizing antibodies, fusion proteins), but can also be accomplished by inhibiting the inflammasome-mediated processing of IL-1β [59,60]. These strategies hold the promise of even an increased therapeutical effect as they can potentially influence not a single cytokine, but a cascade of molecules and downstream signaling processes. As therapeutic efficiency, in almost every case, is a double-edged sword, the prediction of possible side effects becomes even harder. As such, a phase II clinical trial of MCC950, a potent and specific NLRP3 inhibitor, has been discontinued due to hepatic toxicity [61]. In contrast, data obtained from phase I and small phase II trials of another inhibitor (OLT1177) have so far shown good safety profiles [62]. However, to the best of our knowledge, currently no clinical trials are assessing the potential effects of inflammasome inhibition in cancer.

The promiscuity of the ligand receptors of the IL-1 family opens the door for another more widespread treatment approach. The IL-1 receptor accessory protein (IL-1RAcP, also termed IL-1R3), for example, is critically involved in mediating the functions of not only IL-1α and IL-1β, but also of IL-33 and the different IL-36 cytokines (IL-36α, IL-36β, IL-36γ [40]. Treatment with monoclonal antibodies targeting IL-1RAcP can thus inhibit the function of all mentioned cytokines, as recently illustrated in different inflammatory in vivo models (Figure 2) [63]. CAN04, a fully humanized antibody to IL‑1RAcP [64], is currently under investigation for the treatment of different solid tumors (Table 1; NCT03267316, NCT04452214). Figure 2 summarizes the different therapeutic strategies for blockage of IL-1 signaling.

Besides these classical protein-based immunotherapeutic approaches, cellular immunotherapy has developed into a new pillar of cancer therapy, at least in hematological malignancies. In 2018 and 2019, researchers published two studies demonstrating the pivotal function of IL-1 in the development of two of the most common side effects of T cell-based therapies: cytokine release syndrome (CRS) and immune effector cell-associated neurotoxicity (ICANS) [65,66]. For this reason, we will recapitulate the proposed mechanism involved in the development of CRS and ICANS and illustrate the potential of IL-1 blockade in treatment of these diseases.

## 3. IL-1 Blockade in Cellular Therapies 

Chimeric antigen receptor (CAR) T cells targeting CD19 were granted U.S. Food and Drug Administration (FDA) approval in 2017 after demonstrating nearly unprecedented response rates in heavily pretreated patients suffering from acute lymphoblastic leukemia (ALL) or diffuse large B cell lymphoma (DLBCL) [67,68,69,70]. While the overall response rates of axicabatagene ciloleucel (Yescarta^®^) and tisagenlecleucel (Kymriah^®^) reach up to 55% in B cell lymphomas and 93% in ALL [71], the majority of patients develop acute, and in some cases life-threatening, adverse effects. Side effects include on-target off-tumor reactions of the applied CAR T cells aimed at other cells expressing the target antigen. Administration of anti-CD19 CAR T cells, for example, leads to prolonged B cell aplasia [68,72]. Furthermore, anaphylaxis, tumor lysis syndrome or unpredicted cross-reactivity with non-tumor cell antigens can occur [73]. However, as described, the most common side effects of CAR T cells therapy are CRS and to a lesser extent ICANS. However, these side effects are not only observed in CAR T cell therapy, but are common to other T cell activating strategies as well. As such, antibody-based therapies have been shown to induce CRS, as for example observed following administration of the anti-T cell antibody muromonab-CD3 (OKT 3) [74]. New antibody formats such as bispecific T cell-engaging antibodies, for example, targeting CD3 and CD19 (blinatumomab) or CD33 (AMG 330), have also been attributed with the development of CRS [75,76]. In CAR T cell therapy, recent publications reported incidence rates of CRS of nearly up to 100% [77,78,79] and immune effector cell-associated neurotoxicity ICANS of up to 50% [80]. Thus, developing new treatment options for CAR T cell-mediated toxicities has become an important part of immuno-oncological research.

### 3.1. Cytokine Release Syndrome

The term cytokine release syndrome has gained universal attention, as a state of hyper-inflammation has been identified as one of the major contributors to severe courses of SARS-CoV-2 infections [81,82]. Recently, Fajgenbaum and June [83] published an extensive review in the New England Journal of Medicine with the goal of establishing a unifying definition for CRS. This is needed, as some of the currently used definitions are not able to provide a clear delineation between an appropriate inflammatory response and CRS (National Cancer Institute), while others (American Society for Transplantation and Cellular Therapy) only focus on iatrogenic causes of CRS [83].

Following CAR T cell administration, CRS is the most prevalent adverse effect, manifesting usually within one week after T cell transfer. In almost every case, patients develop fever. Additionally, patients present with fatigue, anorexia, headache, rash or diarrhea. In severe cases, patients can rapidly deteriorate, requiring intensive care measures. Common causes for these severe courses include, but are not limited to, disseminated intravascular coagulation, acute respiratory distress syndrome and acute kidney failure [83]. While over the last years, some progress has been made to identify driver cytokines through preclinical models and associated clinical research, the exact underlying mechanisms involved in the development of CRS are unclear. Figure 3 summarizes the currently hypothesized mechanism underlying the development of CRS and or ICANS. Different studies have shown that following infusion of CAR T cells, interactions of tumor cells and T cells lead to an activation of the latter, resulting in the secretion of proinflammatory cytokines such as TNF-α, interferon gamma (IFN-γ) and granulocyte-colony stimulating factor (GM-CSF) [84]. These in turn act on host immune cells, primarily monocytes and macrophages, leading to the secretion of excessive amounts of IL-1, IL-6 and other proinflammatory mediators (e.g., inducible nitric oxide synthase, iNOS). In this regard, IL-1 acts a major inducer of proinflammatory cytokines. Analysis of the kinetics of released proinflammatory cytokines after infusion of CAR T cells, for example, revealed that IL-1 precedes IL-6 release by up to 24 h [65]. Thus, as depicted in Figure 3, it is hypothesized that in CRS, IL-1 amplifies IL-6 signaling, as it has been described previously [85]. Furthermore, besides this cytokine-dependent cross-play between immune cell subsets, direct interaction of CAR T cells and host myeloid cells mediated by the CD40–CD40 ligand (CD40L) axis have been demonstrated (Figure 3). As such, CAR T cells expressing murine CD40L lead to exacerbated CRS symptoms in murine vivo models [66].

Together, these mechanisms induce a hyperinflammatory syndrome with high serum levels of IL-6, TNF-α and other cytokines (e.g., IL-2, IL-8, IL-10), which can result in the development of life-threatening adverse effects [73]. Cytokine profiles comparing CRS to other hyperinflammatory disease states revealed similarities to hemophagocytic lymphohistiocytosis or macrophage-activation syndrome [86]. High serum levels of IL-6 and IFN-γ are able to activate endothelial cells, which leads to the production and secretion of endothelial-derived proinflammatory mediators. As such, in CRS, high serum levels of von Willebrand factor (VWF) and angiopoietin-2 (Ang-2) have been observed [87]. Free Ang-2 can further enhance endothelial activation, creating a positive feedback loop of endothelial activation and leading to increased microvascular permeability (Figure 3) [88]. Furthermore, in vitro studies have shown that IL-6 can downregulate tight junctions on endothelial cells and thus directly disrupt the endothelial integrity (Figure 3) [89]. Autopsy studies of patients who died from severe CRS and ICANS have given histological evidence for these described blood–brain barrier (BBB) dysfunctions [90,91].

### 3.2. Immune Effect Cell-Associated Neurotoxicity Syndrome (ICANS)

As mentioned, another frequently observed adverse effect following CAR T cell infusion is ICANS. Signs of neurotoxicity usually develop within one to three weeks after T cell administration. ICANS can develop concurrently with CRS or in the absence of it. Symptoms are manifold, including but not limited to aphasia, tremor, dysgraphia and lethargy [84].

As for CRS, the regulatory pathways leading to the development of ICANS have yet to be fully identified. While the occurrence of ICANS with both CD19-targeting antibodies (e.g., the bispecific antibody blinatumomab) and anti-CD19-CAR T cells initially raised the question of antigen expression in the central nervous system (CNS), results from newly conducted CAR-T cell trials, for example, targeting B cell maturation antigen (BCMA) or CD22, contradict this hypothesis. Both administration of anti-BCMA or anti-CD22 CAR T cells resulted in the development of ICANS, suggesting rather indirect effects of the applied therapies [92,93]. Importantly, the manifestation of ICANS in the absence of CRS makes it plausible that the underlying molecular pathways between these two diseases differ to at least some extent.

Histopathological analysis of cerebrospinal fluid (CSF) from patients suffering from grade 3 or 4 ICANS have shown significantly higher infiltration of CD14+ monocytes [94]. Thus, monocytes and their effector cytokines have been implicated as playing an essential role in the development of ICANS after administration of CAR T cells [77]. Correlative analysis of proinflammatory mediators elevated in patients with ICANS during the ZUMA-1 trial (NCT02348216) have revealed GM-CSF, ferritin and IL-2 as the markers most strongly associated with neurotoxicity. Along these lines, preclinical studies have helped to highlight the fundamental role of GM-CSF secreted by activated CAR T cells in CAR T cell-mediated neurotoxicity. Treatment with the anti-GM-CSF antibody lenzilumab, for example, was able to reduce the severity of both CRS and ICANS in mouse models, while the effectiveness of CAR T cell therapy was even further improved [80].

### 3.3. Treatment Guidelines of CRS and ICANS 

The severity of these toxicities has led to the establishment of multidisciplinary treatment teams specialized in the diagnosis and treatment of these adverse effects. Access to intensive care units (ICU) and experience in the management of these adverse effects are an important prerequisite for the successful application of treatment [95]. A priori exclusion of comorbidities and close hemodynamic monitoring are also imperative [96].

Treatment options for both diseases need to fulfill certain requirements. Strategies should ideally not interfere with the function of CAR T cells. To this end, treatment with high dosages of corticosteroids was shown to dampen the expansion of CAR T cells with a subsequent reduced antimalignancy effect and a higher frequency of disease recurrence [97]. Tocilizumab, an anti-IL-6 receptor-blocking antibody, does not seem to interfere with the function of CAR T cells, and thus emerged as the first-line treatment option for CRS [98]. Patients responding to tocilizumab treatment exhibit rapid response rates with decrease in respiratory rate, heart rate and CRP levels within 10 h [73]. However, treatment of ICANS differs due to the observed ineffectiveness of IL-6 blocking agents in this disease. Tocilizumab, a monoclonal antibody with an estimated molecular size of around 150 kilodalton, is predicted to not be able to pass through the BBB. With effective receptor blocking only in the periphery, not in the CNS, IL-6-mediated signaling cascades in the CNS can potentially be enhanced. In line with these considerations, deterioration of neurotoxicity-associated symptoms has been reported after application of tocilizumab in the ZUMA-1 trial [84]. Thus, therapeutic options are limited to the administration of antiepileptic drugs such as levetiracetam and of corticosteroids.

### 3.4. Emerging Role of IL-1 Blockade in CRS and ICANS Treatment

As described, current first-line treatment for CRS or ICANS recommend treatment with either tocilizumab for CRS or corticosteroids for ICANS. However, recent preclinical studies have identified IL-1 blockade as a potential promising strategy to mitigate CAR T cell-associated toxicities. After developing a new, highly sophisticated preclinical model for CRS and ICANS, Norelli et al. assessed the potential of either tocilizumab or anakinra for preventing CAR T cell-mediated toxicities [65]. Interestingly, in line with the abovementioned limitations of tocilizumab, only treatment with the IL-1Ra anakinra, not with tocilizumab, protected mice from developing lethal neurotoxicity. Anakinra is able to cross the BBB and is being used for treatment of neurological symptoms caused by hereditary and acquired hyperinflammatory syndromes [99,100,101].

Importantly, IL-1 blocking therapy holds another important promise. So far, preclinical research suggests that anakinra does not seem to hamper CAR T cell function (both proliferation and killing capacity) [65]. Therefore, anakinra could potentially be superior to corticosteroid-based treatment regimes in ICANS. Figure 3 summarizes the hypothesized pathophysiological mechanism of CRS and ICANS and highlights the potential role of IL-1 blockade.

### 3.5. Clinical Validation of IL-1 Targeted Therapy in Combination with CAR T Cell Therapy

Clinical validation of these strategies is already on its way. Preliminary data presented by investigators from the MD Anderson Cancer Center assessed the efficacy of anakinra to mitigate CAR T cell-induced toxicities [102]. While merely being a proof-of-concept study and while the overall number of treated patients was rather low, the researchers observed clinical benefit in four out of six patients. The overall patient outcome in this study was rather dismal; however, given the heavily pretreated patient cohort and an overall refractory disease state of the treated patients, further investigation is obligatory. Thus, results from newly initiated trials will hopefully be able to decipher the effects of IL-1 blockade in CRS and ICANS.

Table 2 gives an overview of currently ongoing clinical trials assessing IL-1 blockade in combination with CAR T cell therapies. Five out of six trials aim to elucidate the effect of anakinra in combination with anti-CD19 CAR T cells either in patients suffering from non-Hodgkin lymphoma (NHL) or ALL. One trial (NCT03430011) examines the combination of anti-BCMA CAR T cells and anakinra in the treatment of MM. In line with treatment guidelines, infusion of CAR T cells is preceded by lymphodepleting chemotherapy. Four of the six conducted trials have a similar treatment schedule: CAR T cells are intravenously administered on day 0, while subcutaneous administration of anakinra is carried out for 5 to 13 days (NCT03430011, NCT04432506, NCT04359784, NCT04150913). One study (NCT04148430) includes a study arm in which anakinra is administered at day 2 or after developing fever, as an indicator of CRS. NCT04205838, conducted by the Johnson Comprehensive Cancer Center at the University of California, will apply anakinra subcutaneously every 6–12 h (total of 12–36 doses) in patients with clinical evidence of ICANS (any grade) or CRS ≥grade 3.

## 4. Conclusions

In spite of substantial efforts from investigators all over the world, the role of IL-1 blocking strategies in cancer therapy is still far from being clear. Several new clinical trials assessing different IL-1 blocking agents have been initiated, and the upcoming years will determine their effectiveness. However, identification of the ideal clinical condition will be of utmost importance. As described by us, but also by other groups, the IL-1 cytokines possess both pro- and antineoplastic properties. Recent reports illustrated the pivotal function of IL-1β for suppressing metastatic colonization [103], with IL-1 blockade leading to increased metastatic burden in mice [104]. To identify and address these divergent effects of IL-1 blockade in cancer therapy, continuing the careful and stepwise preclinical investigation will be just as in important as explorative clinical research. Important future tasks for preclinical researchers will include the illustration of the role of IL-1 in cancer depending on the cellular source, the levels of IL-1 produced and its target. The continuing enhancements in single cell profiling and the resulting development of inter- and intracellular networks inside the tumor hold promise to further unravel the delicate and complex interplay of IL-1 with all the components of the tumor microenvironment. 

Furthermore, it will be of great interest to see whether alternative treatment strategies —either through modulation of the inflammasome-mediated IL-1 processing mechanism or through multi-cytokine targeting approaches—will show clinically relevant response rates and acceptable safety profiles in cancer patients. Finally, results from clinical trials investigating combination treatment of IL-1-blocking agents and CAR T cells will be eagerly awaited to see if promising preclinical data can be translated into clinically relevant treatment options.

## Figures and Tables

**Figure 1 cancers-13-00477-f001:**
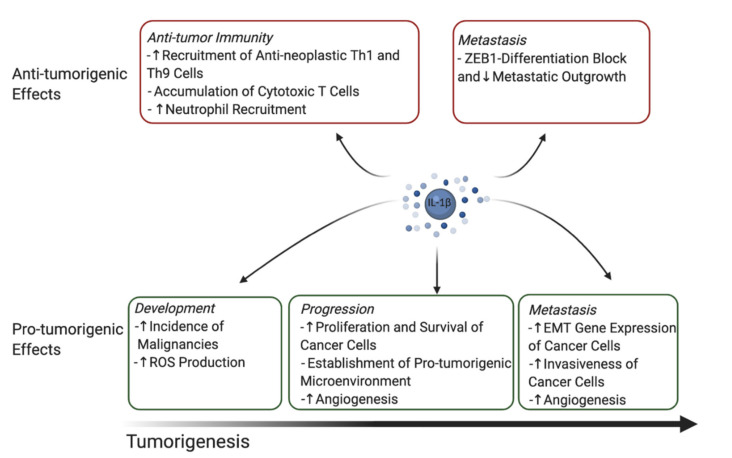
Overview of pro- and antineoplastic functions of IL-1β. Th1/Th9, T helper 1/9 cells; ZEB1, zinc finger E-box-binding homeobox 1; ROS, reactive oxygen species; EMT, epithelial–mesenchymal transition.

**Figure 2 cancers-13-00477-f002:**
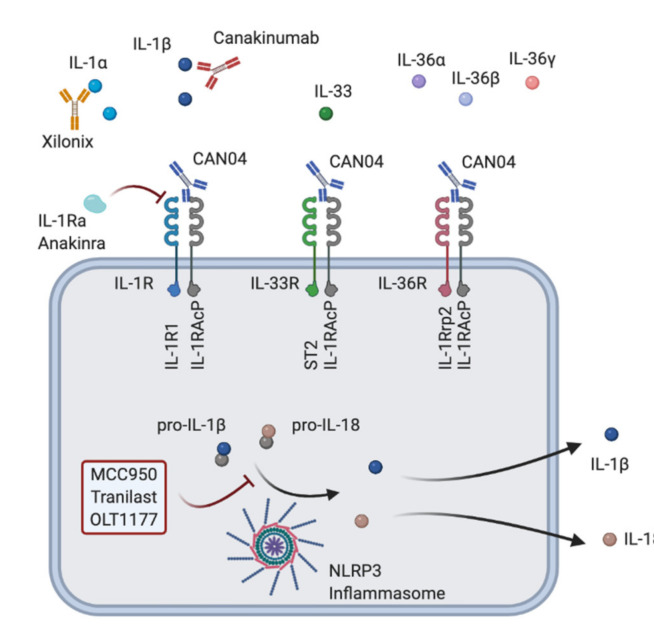
Different treatment strategies to modulate the IL-1–IL-1R pathway. Xilonix, anti-IL-1α antibody; canakinumab, anti-IL-1β antibody, IL-1Ra, IL-1 receptor antagonist; anakinra, recombinant IL-1 receptor antagonist; CAN04, anti-IL-1RAcP antibody; MCC950, inflammasome inhibitor; tranilast, inflammasome inhibitor; OLT1177, inflammasome inhibitor.

**Figure 3 cancers-13-00477-f003:**
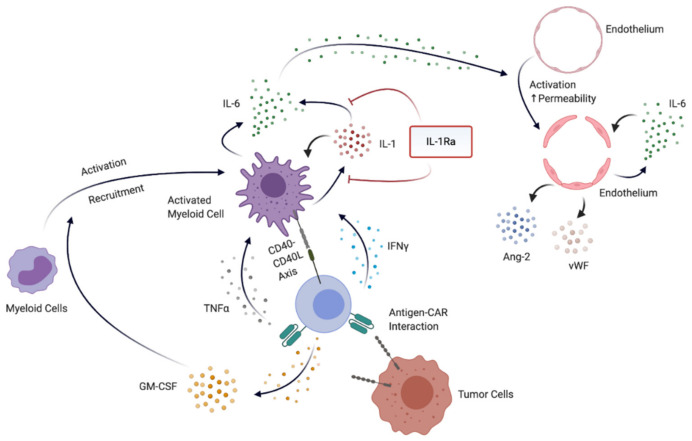
Mechanism leading to CRS and ICANS and potential role of IL-1 blockade. Activated CAR T cells induce cytokine production from myeloid cells through different mechanism: (1) direct interactions between CD40-expressing myeloid cells and CD40L+ T cells or (2) indirect interactions via secretion of proinflammatory cytokines (e.g., GM-CSF, TNFα, IFNγ). Activated myeloid cells produce high amounts of cytokines such as IL-1 and IL-6, which subsequently activate endothelial cells and disrupt vascular integrity. Endothelial cell-derived mediators (e.g., Ang-2 and vWF) can further stimulate endothelial cells, increase permeability and induce coagulopathy. IL-1-receptor antagonists are under investigation to inhibit the IL-1–IL-6 pathway. CRS, cytokine release syndrome; GM-CSF, granulocyte macrophage-colony stimulating factor; ICANS, immune effector cell-associated neurotoxicity; TNFα, tumor necrosis factor alpha, IFNγ, interferon gamma; IL-1Ra, IL-1 receptor antagonist; Ang-2, angiopoietin-2; vWF, von Willebrand factor.

**Table 1 cancers-13-00477-t001:** Overview of clinical trials involving IL-1 blockade in cancer.

Identifier	Conditions	Drug	Status	Phase	Initiated
**IL-1 Receptor Antagonists**					
NCT04121442	Solid Tumors	Isuanakinra + PD-1/ PD-L1 Inhibitor	Recruiting	Phase I/II	2019
NCT02550327	PDAC	Anakinra + Gemcitabine, Cisplatin, Nab-paclitaxcel	Active, not recruiting	Phase I	2015
NCT01624766	Advanced Malignancies	Anakinra + Denosumab, Everolimus	Active, not recruiting	Phase I	2012
**Anti-IL-1β Antibody**					
Hematological Diseases					
NCT04239157	MDS, CMML	Canakinumab	Recruiting	Phase II	2020
Non-Hematological Diseases					
NCT03968419 CANOPY-N	NSCLC	Canakinumab + Pembrolizumab	Recruiting	Phase II	2019
NCT03631199 CANOPY-1	NSCLC	Canakinumab + Pembrolizumab, Chemotherapy	Active, not recruiting	Phase III	2018
NCT03626545 CANOPY-2	NSCLC	Canakinumab + Docetaxel	Active, not recruiting	Phase III	2018
NCT03447769 CANOPY-A	NSCLC	Canakinumab	Recruiting	Phase III	2018
NCT04581343	PDAC	Canakinumab + Spartalizumab, Chemotherapy	Recruiting	Phase I	2020
NCT04028245	RCC	Canakinumab + Spartalizumab	Recruiting	Phase I	2019
NCT03742349	TNBC	Spartalizumab + LAG525, Canakinumab	Recruiting	Phase I	2018
NCT03484923	Melanoma	Spartalizumab + LAG525, Capmatinib, Canakinumab, Ribociclib	Recruiting	Phase II	2018
NCT02900664	CRC TNBC NSCLC	PDR001 + CJM112, EGF816, Canakinumab, Trametinib	Active, not recruiting	Phase I	2016
**Anti-IL-1α Antibody**					
NCT03207724	PDAC	Xilonix + 5-FU, Onivyde	Active, not recruiting	Phase I	2017
**Anti-IL-1R-Antibodies**					
NCT04452214	Solid Tumors	CAN04 + Pembrolizumab	Recruiting	Phase I	2020
NCT03267316	Solid Tumors	CAN04 ± Cisplastin, Gemcitabine, Nab-paclitaxel	Recruiting	Phase I/II	2017

Information was obtained from www.clinicaltrials.gov [52]. Terms used: IL-1 blockade, IL-1, IL-1α, IL-1β, cancer, anakinra, canakinumab, xilonix, NLRP3 inhibitor, inflammasome. PDAC, pancreatic ductal adenocarcinoma; CRC, colorectal carcinoma; SMM, smoldering multiple myeloma; IMM, indolent multiple myeloma; MM, multiple myeloma; MDS, myelodysplastic syndrome; CMML, chronic myelomonocytic leukemia; NSCLC, non-small cell lung cancer; TNBC, triple-negative breast cancer; RCC, renal cell carcinoma; PD-1, programmed cell death protein 1; PD-L1, programmed cell death ligand 1; PDR001, anti-PD-1 antibody; CJM112, anti-IL-17A antibody; EGF816, EGFR-inhibitor; trametinib, MEK-inhibitor; LAG525, anti-LAG-3 antibody; capmatinib, MET-inhibitor; ribociclib, CDK4/6-inhibitor, MABp1, monoclonal anti-IL-1α antibody; onivyde, liposomal irinotecan; CAN04, humanized anti-IL-1RAcP antibody.

**Table 2 cancers-13-00477-t002:** Overview of clinical trials combining IL-1-blockade and CAR T cell therapies.

Identifier	Conditions	Drug	Status	Phase	Initiated
**IL-1-Receptor Antagonist**					
NCT04432506	BCL	Axicabtagene Ciloleucel + Anakinra	Recruiting	Phase I/II	2020
NCT04359784	B-NHL	Axicabtagene Ciloleucel + Anakinra	Not yet recruiting	Phase I/II	2020
NCT04150913	B-NHL	Axicabtagene Ciloleucel + Anakinra	Recruiting	Phase II	2019
NCT04148430	B-ALL BCL B-NHL	Anti-CD19 CAR T Cells + Anakinra	Recruiting	Phase II	2019
NCT04205838	BCL	Axicabtagene Ciloleucel + Anakinra	Recruiting	Phase II	2019
NCT03430011	MM	JCARH125 + Anakinra	Recruiting	Phase I/ II	2018

Information was obtained from www.clinicaltrials.gov [52]. Terms used: IL-1 blockade, IL-1, IL-1α, IL-1β, cancer, anakinra, canakinumab, xilonix, NLRP3 inhibitor, inflammasome. BCL, B cell lymphoma; B-NHL, B cell non-Hodgkin lymphoma; B-ALL, B cell acute lymphoblastic leukemia; MM, multiple myeloma; JCARH125, anti-BCMA CAR T cells.

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
