# Peer review of "Therapeutic Strategies for Targeting IL-1 in Cancer"

_cancers, 2021, doi:10.3390/cancers13030477_

Round 1

Reviewer 1 Report

In this review, Gottschlich et al. summarize appropriately and completely the state of the art on the therapeutic strategies capable of targeting IL-1 in the field of immuno-oncology including new strategies to enhance the safety profile and the efficacy of adoptive T cell therapies. The review is well organized and discursive. Very minor comments should be addressed:

  • The authors should double check the abbreviations, be consistent how they are written and write in full length the first time used.
  • In row 98: modify “lenalidomid” in lenalidomide.
  • In figure 1, I would suggest to write the title in bold and include in the legend the abbreviations used.
  • Row 245: modify “tumor” with tumors.
  • In the beginning of paragraph 3.1, it would be important to include a refence on cytokine release syndrome and monoclonal antibodies with particular reference to blinatumomab.
  • Please insert the NCT of the ZUMA-1 trial in row 336.

Author Response

Response to Reviewer #1

  1. The authors should double check the abbreviations, be consistent how they are written and write in full length the first time used.

We appreciate this comment, which will help to improve the readability of this manuscript. We have inserted appropriate abbreviations wherever necessary and included a list of all used abbreviations in the figure legend 1 and 2.

  1. In row 98: modify “lenalidomid” in lenalidomide.

We have corrected this spelling mistake (Row 99).

  1. In figure 1, I would suggest to write the title in bold and include in the legend the abbreviations used.

We have made the requested changes to the title and, as mentioned above, have included all abbreviations used in the figures in their respective legends (Row 51 – 52).

  1. Row 245: modify “tumor” with tumors.

We have corrected this spelling mistake (Row 264).

  1. In the beginning of paragraph 3.1, it would be important to include a reference on cytokine release syndrome and monoclonal antibodies with particular reference to blinatumomab.

We are grateful for this important comment and have added a section describing cytokine release syndrome as an important adverse effect following antibody-based therapies, with special focus on the newly developing field of T cell activating antibodies (Row 294 – 301).

  1. Please insert the NCT of the ZUMA-1 trial in row 336.

We have inserted the NCT reference number of the ZUMA-1 trial as recommended (Row 372).

Reviewer 2 Report

This review summarizes the role of IL-1 in malignancy and explores its role in CAR-T toxicity and efficacy. It is overall well-written. Both pro- and anti-tumor activities are described.

Comments:

1  Under hematologic malignancies, would mention and develop the work that has been done with IRAK1 inhibitors n leukemia.  The role of the inflammasome as related to IL-1 could also be developed in the area of MDS.

2 .Page 3; line 117; define CRC the first time it is used. That is also true for some other abbreviations (MDSC/TAMS).

3.  Page 4, line 161; what does "to neither" mean?

4. At the bottom of page 7, would eliminate the phrase "just a few weeks ago" as that will no longer be true when this is published.

5. Page 8, lie 310 and 312, would elaborate on serum levels of what; ? IL-6, INF-gamma, other cytokines and chemokines? How does IL-1 interact with those, and what are the implications for inhibiting these simultaneously? Figure 3 helps somewhat in this regard but could be more developed in the text.

6. Page 9, line 367, mediate should be mitigate

7.  Page 10, line 399 should obliagate be obligatory or necessary?

8. Page 10, line 403, define ACT.

9  In the conclusions or elsewhere, it may be good to comment on how IL-1 inhibition effects on pro-vs anti-tumor manifestations can be sorted out

Author Response

Reply to Reviewer #2

  1. Under hematologic malignancies, would mention and develop the work that has been done with IRAK1 inhibitors n leukemia.  The role of the inflammasome as related to IL-1 could also be developed in the area of MDS.

This is a very insightful comment, addressing innovative developments in the field of targeted therapies. We have added a paragraph elaborating recent advances using protein kinase inhibitors to target constitutive signaling downstream of the IL-1 and toll-like receptor pathways in hematological malignancies (Row 100 – 114).

  1. Page 3; line 117; define CRC the first time it is used. That is also true for some other abbreviations (MDSC/TAMS).

We have included appropriate abbreviations and as described in the response to comment 1 of Reviewer #1 have improved overall on the use of abbreviations in the manuscript.

  1. Page 4, line 161; what does "to neither" mean?

We have deleted “to neither” which improved the readability of this sentence.

  1. At the bottom of page 7, would eliminate the phrase "just a few weeks ago" as that will no longer be true when this is published.

We have eliminated “just a few weeks ago” and used the word “recently” instead (Row 307).

  1. Page 8, line 310 and 312, would elaborate on serum levels of what; ? IL-6, INF-gamma, other cytokines and chemokines? How does IL-1 interact with those, and what are the implications for inhibiting these simultaneously? Figure 3 helps somewhat in this regard but could be more developed in the text.

We have rephrased and specified this paragraph (Row 328 – 336) in regards to the IL-1 mediated interactions. Furthermore, we have developed Figure 3 more into the text (Row 321, 322; 331) to improve the understanding of the proposed mechanism of cytokine release syndrome and immune effector cell-associated neurotoxicity.

  1. Page 9, line 367, mediate should be mitigate

We have corrected this spelling mistake (Row 403).

  1. Page 10, line 399 should obliagate be obligatory or necessary?

We have corrected this spelling mistake (Row 435).

  1. Page 10, line 403, define ACT.

We have defined ACT, in this case focusing on CAR T cell therapy (Row 439).

  1. In the conclusions or elsewhere, it may be good to comment on how IL-1 inhibition effects on pro-vs anti-tumor manifestations can be sorted out

In the conclusion we have added a paragraph of possible means, both preclinical and clinical, to unravel the complex functions of IL-1 in the context of cancer treatment (Row 466 – 473). We have based our arguments on the differing role of IL-1 depending on the cellular source, the overall level of IL-1 and its target. We hold the opinion that recent advances in single cell profiling and improvements of stepwise modelling of important cascades in cancer progression will shed light onto these complex functions and will identify a suitable setting for IL-1 targeted therapies in clinical treatment.